# Linguistic based emotion analysis using softmax over time attention mechanism

**Megha Roshan[1], Mukul Rawat[2], Karan Aryan[3], Elena Lyakso[4], A. Mary Mekala[1], Nersisson Ruban[2]***

1 School of Computer Science and Engineering, Vellore Institute Technology, Vellore, India, 2 School of Electrical Engineering, Vellore Institute Technology, Vellore, India, 3 Department of Computer Engineering, Army Institute of Technology, Pune, India, 4 The Child Speech Research Group, St. Petersburg State University, St. Petersburg, Russia

* nruban@vit.ac.in

**Data Availability Statement:** Data and Code Availability Statement We have used two openly available datasets namely Multimodal Emotion Lines Dataset (MELD), International Survey on Emotion Antecedents and Reactions (ISEAR), and

## Abstract

Recognizing the real emotion of humans is considered the most essential task for any customer feedback or medical applications. There are many methods available to recognize the type of emotion from speech signal by extracting frequency, pitch, and other dominant features. These features are used to train various models to auto-detect various human emotions. We cannot completely rely on the features of speech signals to detect the emotion, for instance, a customer is angry but still, he is speaking at a low voice (frequency components) which will eventually lead to wrong predictions. Even a video-based emotion detection system can be fooled by false facial expressions for various emotions. To rectify this issue, we need to make a parallel model that will train on textual data and make predictions based on the words present in the text. The model will then classify the type of emotions using more comprehensive information, thus making it a more robust model. To address this issue, we have tested four text-based classification models to classify the emotions of a customer. We examined the text-based models and compared their results which showed that the modified Encoder decoder model with attention mechanism trained on textual data achieved an accuracy of 93.5%. This research highlights the pressing need for more robust emotion recognition systems and underscores the potential of transfer models with attention mechanisms to significantly improve feedback management processes and the medical applications.

## 1. Introduction

Experimentation for detecting human emotions was first done by Ekman and Friesen in 1978. Since then, several advances have been made in this field and it has become an essential task for many applications. Many algorithms have been developed to detect human emotions and until now, most of the work has been conducted on automating the recognition of facial expressions from video, spoken expressions from audio, and written expressions from the text. This paper describes different models made for recognizing emotions from the textual content of the speech signals. The main objective is to propose a model that can give better inference

an emotional search engine "We Feel Fine'. The data used in this research is openly available, the URLs required to download the data are given below. • Multimodal Emotion Lines Dataset (MELD) : https://web.eecs.umich.edu/~mihalcea/downloads/MELD.Raw.tar.gz • International Survey on Emotion Antecedents and Reactions (ISEAR) : https://www.unige.ch/cisa/research/materials-and-online-research/research-material/ • We Feel Fine: http://www.wefeelfine.org/ The code is openly available with the sample data in Kaggle: https://www.kaggle.com/code/mukulem/notebookbcf0cec82c/notebook.

**Funding:** The authors thank Vellore Institute of Technology, Vellore for providing 'VIT SEED GRANT' for carrying out the initial study related to this research work. The recipient of the disclosed fund is the corresponding author RN. The funders had no role in study design, data collection and analysis, decision to publish, or preparation of the manuscript.

**Competing interests:** The authors have declared that no competing interests exist.

on both textual and audio data as input. The research is divided into two parts. The first part focuses on deriving Mel Frequency Cepstrum Coefficients (MFCC) spectrogram images from speech signals, and then applying deep learning models on those derived images for classification of emotions. But relying on MFCC voice feature coefficients and components of speech signals for emotion classification is not sufficient which varies from person to person. In addition to this, the speech is converted to text format and then the emotions are classified using the words in the textual content. This paper describes another approach where different deep learning-based language models are built and trained on textual data to classify the emotions. Converting the words in the statement into their dense vector layer is the basis for the attention mechanism and incorporating it into the encoder decoder concept to get a better performance is the major technical milestone proposed in this research work. The proposed approach is depicted in Fig 1. In the block diagram, Voice Signal Input is the starting point where the voice signal is received. In the preprocessing stage the voice signal undergoes noise removal and enhancement.

Branch 1—Voice Processing: The preprocessed voice signal is converted into a mel power spectrogram. The spectrogram is then used for emotion classification through a deep convolutional network.

Branch 2—Text Conversion and Analysis: Parallel to the first branch, the voice is also converted into text. The converted text is then sent for preprocessing. An LSTM language model processes the text for sentiment analysis (classifying it as positive or negative) and emotion classification.

A lot of research is focused on extracting the desired features and preparing a suitable model. In the previous trials, a real-time implementation of an emotion recognition system was proposed [1]. The analysis of emotion was done using Facial Expressions, Speech, and

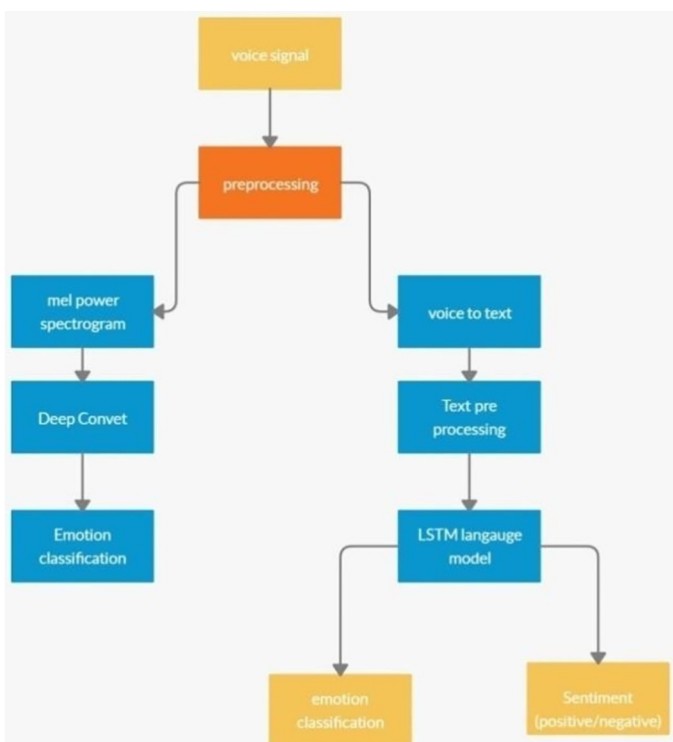

**Fig 1. Flow chart of the proposed model.**

Multimodal Information. Facial Actions, Expressive faces, body gestures, and Speech were taken into consideration for implementing the recognition system. An efficient approach was used to create a face and emotion feature database which was then used for face and emotion recognition [2]. Another approach uses a TensorFlow-based deep learning model for emotion recognition using facial images of seven different categories [3]. The categories of emotions were Angry, Disgust, Fear, Happy, Sad, Surprise, and Neutral. A Speech-based Emotion Recognition was developed in which emotions like anger, happiness, and sadness, were classified using three feature vectors. Pitch, Mel frequency cepstral coefficients, Short Term Energy were the three feature vectors extracted from audio signals [4]. Bakariya, B. et al., used the CNN based deep learning technique to identify the six facial expressions: neutral, sad, joy, anger surprise and joy. As a result, the current human emotion is predicted and the relevant music is recommended. The dataset considered for this study is OAHEGA and FER-2013s and the proposed model gave an accuracy of 73.02% [5]. Another process was used to determine the MFCC coefficients and check their differences to differentiate the emotions [6]. Emotion recognition from audio, dimensional, and discrete categorization using Convolutional Neural Network (CNN) approach was also proposed [7]. A 2D quadrant predictor CNN model was built resulting in 76.2% accuracy by using two different CNN architectures on the RAVDESS dataset. A Multi- Task Learning (MTL) was proposed and the use of gender and naturalness as auxiliary tasks in deep neural networks was explored to improve the generalization capabilities of the emotion models [8]. A complete literature review of speech emotion recognition was done by Babak Basharirad, and Mohammadreza Moradhaseli [9].Various speech and emotional recognition systems-based approaches were discussed and reviewed. Fuzzy Logic was used for Textual Emotion Detection by S. B. Sadkhan and A. D. Radhi [10]. Meena, G. et al., proposed a solution built on deep Convolutional Neural Network model to predict the face sentiment analysis. The datasets FER-2013 and extended Cohn Kanade were used for this study and obtained an accuracy result of 79% and 95% individually [11]. Meena, G. et al., suggested a deep CNN Inception-v3 technique, which mainly focuses on the human face to detect and classify the emotions. The datasets used for this study attains 99.5% accuracy for CK+, 86% for JAFFE, and 73% for FER2013 [12, 13]. Meena, G. et al., proposed a transfer learning model to classify image-based sentiment. The hyper parameters in the techniques such as VGG-19, Inception-v3, and XceptionNet were adapted and gave a better result compared to the existing model [14]. Emotion detection and sentiment analysis were done on textual data and techniques like fuzzy logic and neural network systems to separate emotions from text present in different sites were proposed. F. Mozafari and H. Tahayori [15], proposed a method for detecting and analyzing emotions based on similarity using Vector Space Model (VSM) and Short text Algorithm for Semantic similarity measure known as STASIS methods. They proposed three methods of VSM, keyword base, and STASIS with considering maximum and average of similarity in each emotion category to detect emotions in text. The proposed work aims at recognizing the emotion of human speech through deep learning algorithms. The initial step deals with preprocessing of the Multimodal Emotion Lines Dataset, fondly referred to as MELD. Text preprocessing involving lowercasing, removing punctuation and digits, removing redundancy, and spelling correction that provide us with a cleaner dataset to work with. Furthermore, stemming, amortization, and normalization techniques are incorporated to achieve desirable data form. This data is fed to various deep learning models for classification. The technique which involves the extraction of prosodic and spectral features of an acoustic signal like pitch, energy, formant, Mel-frequency Cepstrum Coefficients (MFCC) and Linear Prediction Cepstral Coefficient (LPCC) along with their classification using a cubic spine Support Vector Machine (SVM) classifier model is proposed in [16]. The models are then compared based on accuracy and error loss. The algorithms used are refined and well thought over

to increase the accuracy and limit the loss due to error to cater to the efficient prediction results.

The flow of the paper is in the following order. Dataset and data preprocessing are mentioned in section 2 referred to as materials and methodology. The various classification algorithms are depicted in section 3. Results and predictions are discussed in section 4 followed by a conclusion to provide a better understanding of the future development and extension of the project. To obtain a final emotion label, we combines the outcomes from both modalities, taking into account their respective confidence scores and output the resulted emotion which has highest confidence score. The mechanism of choosing the highest confidence level is used as the voting scheme in the proposed approach. This multimodal approach enhances the overall accuracy and reliability of our emotion classification system by harnessing complementary information from both voice and text sources.

## 2. Materials and methods

Three different sources for dataset are used in this study: Multimodal Emotion Lines Dataset (MELD), International Survey on Emotion Antecedents and Reactions (ISEAR), and an emotional search engine named We feel fine.

To rigorously evaluate the performance of our emotion recognition models trained on the MELD, ISEAR, and We Feel Fine datasets. We designed a versatile simulation environment having Dataset from different diversity, culture, age groups as well as gender. This simulation framework allowed us to replicate real-world scenarios, introducing controlled variations in cultural context, linguistic diversity, and emotional expressions. By simulating diverse conversational dynamics and cross-cultural variations, we assessed the adaptability of our models to different contexts. This simulation approach played a pivotal role in uncovering potential limitations, biases, and challenges that may arise due to cultural, linguistic, and contextual differences, ensuring the robustness and applicability of our models across various real-world scenarios. Biasness can be improved by fine tuning the same model with a new dataset in different simulation environment.

### 2.1. Dataset

Three different well-documented datasets are used for this study, all the datasets are available openly and free to download from the mentioned sources.

**2.1.1. MELD.** A Multimodal Multi-Party Dataset for Emotion Recognition in Conversation named Multimodal Emotion Lines Dataset (MELD) which is an extension of Emotion Lines Dataset has been used to train the model [17]. It contains 1433 dialogues and 13000 utterances extracted from various episodes of the famous American sitcom series "Friends". Each utterance in the dataset incorporates three modalities- text-based, audio-based, and visual-based. The utterances in each dialogue are annotated with the emotion and sentiment associated along with the Dialogue_ID and Utternace_ID. It also mentions the season and episode with the StartTime and EndTime of the utterance. Table 1 shows the MELD dataset format and Fig 2 shows the MELD data distribution.

This dataset uses information from multi-parallel data channels to make decisions. The advantage of selecting Multimodal Emotion Lines Dataset (MELD) is that it incorporates multi-party conversations, unlike the rest that rely on a dyadic conversation that limits the scalability of emotion recognition to the conversation between two people. MELD is believed to have improved context modelling for better performance at emotion recognition, as it is based on multimodal data sources. Other multimodal conversational datasets including IEMOCAP and SEMAINE are dyadic thus generating the need to use MELD.

**Table 1. MELD dataset format [17].**

| Sr No. | Utterance | Speaker | Emotion | Sentiment | Dialogue _ID | Utterance _ID | Season | Episode | Start Time | End Time |
|---|---|---|---|---|---|---|---|---|---|---|
| 1 | also I was the point person on my company's transition from the KL-5 to GR-6 system. | Chandler | neutral | Neutral | 0 | 0 | 8 | 21 | 00:16:16,059 | 00:16:21,731 |
| 2 | You must've had your hands full. | The Interviewer | neutral | Neutral | 0 | 1 | 8 | 21 | 00:16:21,940 | 00:16:23,442 |
| 3 | That I did. That I did. | Chandler | neutral | Neutral | 0 | 2 | 8 | 21 | 00:16:23,442 | 00:16:26,389 |
| 4 | So let's talk a little bit about your duties. | The Interviewer | neutral | Neutral | 0 | 3 | 8 | 21 | 00:16:26,820 | 00:16:29,572 |
| 5 | My duties? All right. | Chandler | surprise | positive | 0 | 4 | 8 | 21 | 00:16:34,452 | 00:16:40,917 |

**2.1.2. ISEAR.** International Survey on Emotion Antecedents and Reactions (ISEAR) covers the statements of students, psychologists, and non-psychologists on their experience regarding the 7 major emotions comprising of joy, fear, anger, disgust, shame, sadness, and guilt [18]. The ISEAR dataset distribution is shown in Fig 3. The dataset collection project,

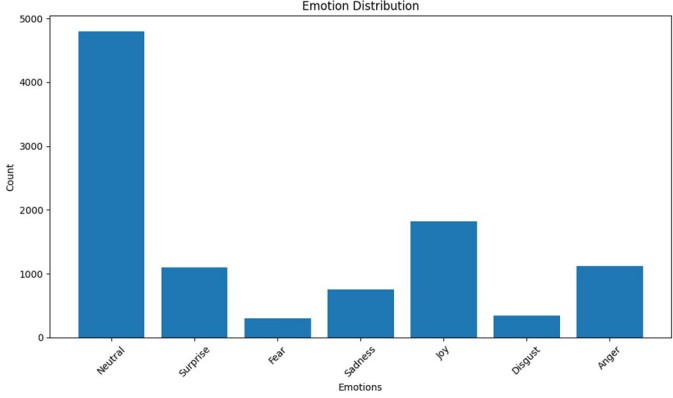

**Fig 2. MELD dataset distribution.**

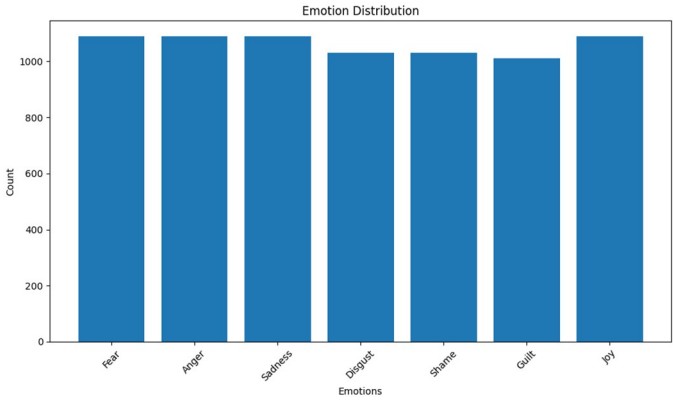

**Fig 3. ISEAR dataset distribution.**

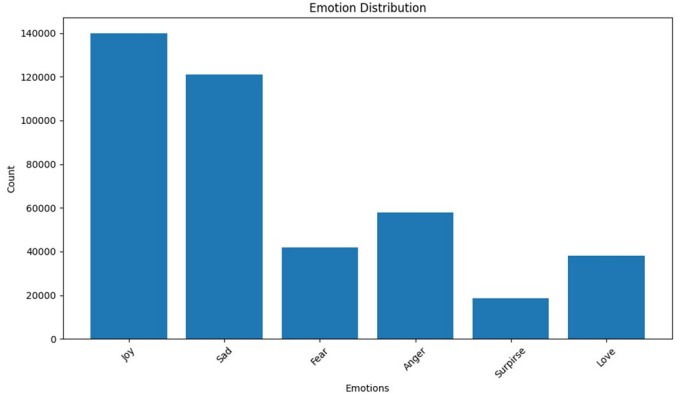

**Fig 4. We feel fine dataset distribution.**

under the supervision of Klaus R. Scherer and Harald Wallbott, has statements that deals with how the respondents encountered the situation and what was their reaction to it.

The dataset contains fully covered views of around 3000 people located across 37 countries spanning 5 continents.

**2.1.3. We feel fine.** We feel fine is an emotional search engine that harvests human emotion from web blogs, social networking sites [19]. It extracts the sentences which include "I feel" and "I am feeling" as well as the age, gender, and location of the people who are the authors of those sentences. The we feel fine dataset distribution is shown in Fig 4.

## 2.2. Data preprocessing

Raw input texts are not ideal to train the deep learning model. Cleaning of text in text preprocessing is an essential step in bringing the text into a format that is predictable for proposed work. The quality of the input data will have a direct influence on the performance of the model. There are several methods of preprocessing but choosing steps that are in line with the proposed task is more important. Text pre-processing is done in python by using libraries such as NLTK, Regex, and basic Python string operations. The first step of pre-processing is lowercasing, converting letters to Lowercase is one of the simplest as well as effective preprocessing techniques. It eventually helps in reducing the size of the vocabulary in our text data. The words which have the same meaning but have different cases are all converted to lowercase. For example "Hate" and "hate" have the same meaning but will be counted as different words due to different cases, hence lowercasing will solve these problems. The next step is removing punctuation (!"#$%&'()*+,-./:; < = >?@[\]^_'{|}~) from the dataset. For the emotion analysis, punctuation does not provide any valuable information, hence removing them from the corpus will increase productivity and effective training of the models. Similarly, digits also do not influence emotion analysis and thus are also removed from the corpus.

Stop words are a set of words that are commonly used in the language such as "a"," the"," there"," are", etc. Again the main intuition behind removing stop words is to remove words that have low information and do not help in distinguishing the type of emotion. Removing such words reduces the amount of unwanted data.

For example "i hate you" is classified as anger. "Hate" is the only important word here that is responsible for anger emotion. The words "I" and "you" are stop words with no significance in detecting the emotion. After removing stop words, next comes duplicates or repetitive of

**Table 2. Example for stemming.**

| S. No | Original word | Stemmed word |
| --- | --- | --- |
| 1 | happiness | happy |
| 2 | happily | happy |
| 3 | Sadly | Sad |

the same word in the dataset. For example in this sentence from the corpus "Water breaking, what do you mean? What's that, water breaking?" water breaking is repeated twice. Such duplicates are removed from the corpus to reduce the data size for efficient training.

Spelling correction is the process of correcting the spelling of the word. In the corpus, some words such as "awwwwesome" where the person seems to be excited are misspelled. Correction of spelling is done using the pyspellchecker library in python which loops through each word and corrects the spelling. Example changing "awwwwesome" to "awesome".

Next comes stemming and few sample words are given in Table 2. Stemming is the process of transforming each word to its root word. The root word here is the canonical form of the original word. It uses a crud heuristic algorithm by chopping the words from the end to convert them into their root word. There are many methods available for stemming. In this paper, the **Porters algorithm** which is the most effective algorithm for English words is applied using the NLTK library in python.

After stemming next step is lemmatization which is closer to stemming but the only difference is that lemmatization does not chop off the word but it links the word with the same meaning to one word by morphological analysis of words. Last and the most important step in pre-processing is Normalization which is the process of transforming the word into its standard form. Dataset consists of conversational data, hence it is noisy as it has various abbreviations, misspelled words that may be left out in spelling corrections, and words that are out of vocabulary. Such words may have a great influence on emotion analysis and it improves the overall quality of the dataset. Normalization is done using normalize and spacy library in python.

Aiming for better accuracy with imbalanced data is counterproductive because the deep learning model may not have the ability to handle imbalanced data. Imbalance data means one or more class is higher than the other.

Common emotions from each of the datasets are used to the built final dataset and balancing each emotion value. The data set distribution after balancing is shown in Fig 5.

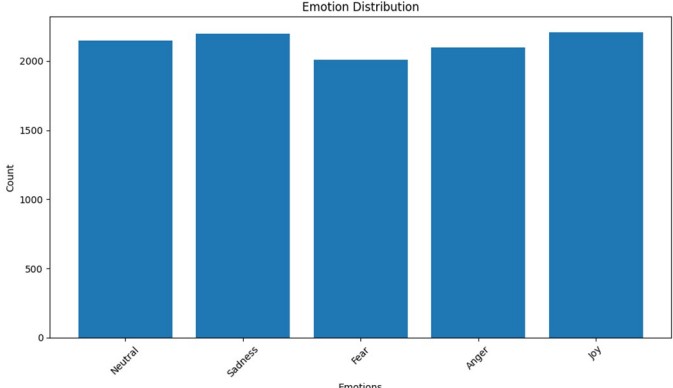

**Fig 5. Dataset distribution after balancing.**

## 2.3. Algorithm

The overall implementation of the work is presented as a step by step process below:

Stage 1: Preprocessing and data preparation

Step 1: Read the text data with a specific sequence length

Step 2: Apply preprocessing steps described in data preprocessing

Step 3: Construct embedding matrix of word ID to corresponding matrix and word ID to word

Step 3: Transform text data into an embedding vector of 200 dimensions

Stage 2: Attention Model

Step 1: Build Encoder and Decoder LSTM layer of size equal to the max length of the sequence

Step 2: Build softmax over time layer.

Step 3: Feed the Embedding matrix of size Max length x 200 to the encoder layer

Step 4: Pass the output of decoder layer to the final softmax layer

Stage 3: Output prediction

Step1: Output from the softmax layer is fed to word ID to word mapper

Step 2: Interpret the emotion

Softmax function independently applies to each set of logits. This converts the logits into a probability distribution for each emotion category at each time step. This means that every word or token in the text will have a probability distribution over emotions.

# 3. Classification models

This research initiative is focused on deep learning in natural language processing based models. Models are categorized into two categories Text classification and sequence to sequence.

## 3.1. Text classification

Extracting values from unstructured data can be expensive and tedious. Text classification can help structure the data by categorizing texts into various pre-defined tags or categories based on its context. We explored few advanced techniques to achieve the same. The techniques are discussed below:

**3.1.1. Long short-term memory (LSTM).** Sequence classification is a predictive modeling problem where you have some sequence of inputs over space or time and the task is to predict a category for the sequence [20]. LSTM is an extended form of Recurrent Neural Network (RNN) with the addition of an Internal Memory Cell. Sequence classification is a predictive modeling problem where you have some sequence of inputs over space or time and the task is to predict a category for the sequence. In this paper LSTM recurrent neural network models are built for sequence classification in Python using the Keras deep learning library. The LSTM architecture is shown in Fig 6.

LSTM networks can regulate the flow of outgoing and incoming data through a gate structure which enables it to learn when to ignore a current input or remember the past hidden state $h_{t-1}$. A total of three gates are used in a basic cell of an LSTM network. In the first step, the forget gate determines how much information previous cell state should be discarded or

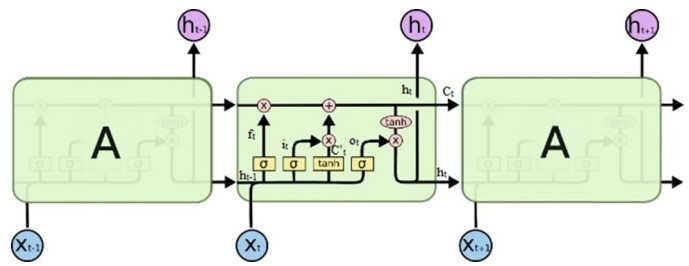

**Fig 6. LSTM architecture [20].**

kept. In the second step, the input gate determines the new information that is to be stored in the current cell state. Eqs 1 and 2 are used to calculate forget gate $f_t$ and input gate $i_t$.

$$f_t = \sigma\left(W_f.[h_{t-1}, x_t] + b_f\right) \tag{1}$$

$$i_t = \sigma\left(W_i.[h_{t-1}, x_t] + b_f\right) \tag{2}$$

And then the updated value of the cell state is calculated using the results achieved from the input gate and forget gate and in the final step, the output gate determines the output which influences the whole network. In Eq 3, a tanh layer creates a new proposed cell $C'_t$

$$C'_t = tanhtanh(W_C.[h_{t-1}, x_t] + b_C) \tag{3}$$

In Eq 4, updating the value of old state, $C_{t-1}$, into the new cell state $C_t$.

$$C_t = f_t * C_{t-1} + i_t * C'_t \tag{4}$$

Multiplying the old state by $f_t$, forgetting the things we decided to forget earlier. Then adding $i_t * C'_t$. Eq 5 determines the value of output gate

$$o_t = \sigma(W_o.[h_{t-1}, x_t] + b_o) \tag{5}$$

and finally the hidden state $h_t$ is calculated using (6).

$$h_t = o_t * tanhtanh(C_t) \tag{6}$$

*3.1.1.1. Training data Preparation and Modeling.* In LSTM model, to make the neural model understand the word, the sequences are converted into a set of integers by tokenizing and then padding to the maximum length which is set to be 50. So that the length of the input sequences are equal. Output labels that are also categorical are converted into numbers and one-hot encoded.

For example, in the sentence "I am feeling really bitter", after tokenizing the sentence output is [1–5] where each integer is mapped with its corresponding word. Now after applying padding to the maximum length of 10 output is [1,2,3,4,5,0,0,0,0,0] where "0" corresponds to the padded token. After preparing data, the dataset is now split into a training and validation set of 80:20.

In an LSTM model, the first layer consists of an embedding layer which has 50 neurons corresponding to the maximum length of the sentences. Then comes two LSTM layers of 256 and 128 memory units respectively followed by a dropout layer adding a dropout of 40% in layers. After dropout layer two dense fully connected layers are implemented, 1st layer has 128

neurons with ReLu activation and the next layer is the classification layer with 5 neurons with softmax activation.

**3.1.2. Bidirectional Encoder Representations from Transformers (BERT).** BERT relies on the transformer architecture that has been pre-trained on a humongous corpus containing unlabeled text which incorporates the Book corpus and Wikipedia figuring to around 3,100 million words roughly [21]. In addition to this, BERT follows a Bi-directional approach implying that, during the training phase, the model learns both from the left and the right side of the token. It came out as an efficient solution to polysemy. This helps BERT in generating a deeper insight into the sentence. The advantage of selecting BERT lies in the flexibility to fine-tune it by introducing a few task-specific output layers [22].

BERT is available in two variants:

BERT Base: 12 layers (transformer blocks), 12 attention heads, and 110 million parameters

BERT Large: 24 layers (transformer blocks), 16 attention heads and, 340 million parameters

Transformers include two mechanisms: encoder and decoder. Since BERT was established to generate a language model, it uses only the encoder mechanism.

It follows two modifications to ensure bi-directionality along with pre-training on large corpus: Masked Language Model (MLM) and Next Sentence Prediction.

The MLM BERT is depicted in Fig 7. MLM is based on the idea to randomly mask 15% of the words in the dataset. Masking a word refers to replacing the word in the sentence with the [MASK] token. To further ensure that efficiency keeping in mind that the [MASK] token does not appear in fine-tuning, 80% of the time, masked words were replaced with [MASK], 10% of the times the words were replaced with a new random word, and another 10% of the time, the sentences were left unchanged.

Next Sentence prediction is introduced to cater to the relation between sentences and it is shown in Fig 8. Out of the total number of pairs of sentences taken from the training data, half of the pairs have the second sentence as the successor of the first sentence and thus are labeled as "IsNext". For the remaining half of the pairs, the second sentence is any random sentence from the corpus that has no relation to the first sentence and is labeled as "NotNext".

*3.1.2.1. Training data Preparation and Modeling.* Input data for BERT has to be in a specific format, with some special tokens such as [CLS] token which is added at the beginning of sentence and [SEP] end of sentence token added at the end of the sentence. After adding tokens all the words in the dataset has to be tokenized according to the BERT's vocabulary. For each

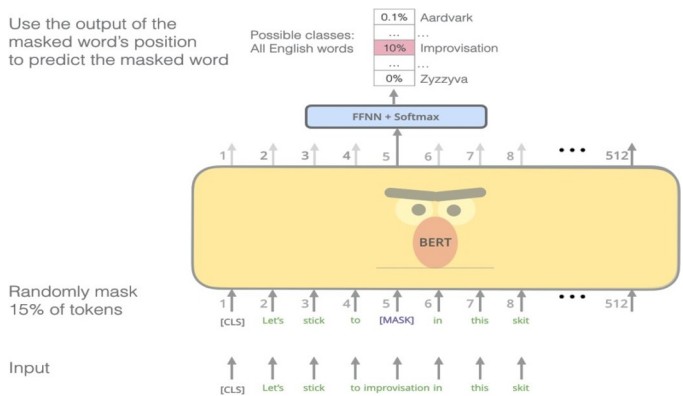

**Fig 7. MLM BERT [22].**

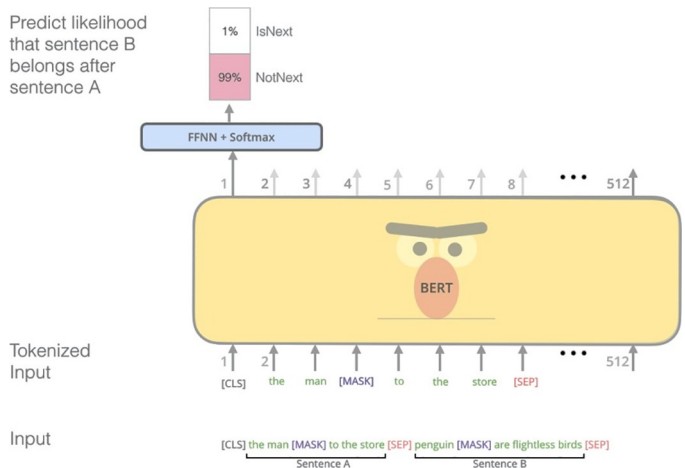

**Fig 8. Next sentence prediction BERT [22].**

tokenized sentence, BERT needs a sequence of integers that is input ids to identify each input token map to its index number in vocabulary.

After tokenizing, dataset is divided into 80:20 training and validation ratio. BERT for Sequence Classification, which is a BERT pre-trained model, is fine-tuned with a single layer on top for classification.

## 3.2. Sequence to sequence

Sequence to sequence is a training model that takes in a sequence of items. These items can be words, time series, letters, etc. The output is a translated or converted sequence of items. In this work, few sequence to sequence models are explored and it is listed below:

**3.2.1. Encoder decoder model.** Sequence to Sequence model finds its application in the wide domain of problems that are based on Machine translation, Video Captioning, and Speech Recognition, particularly the model being used to generate its best performance by overcoming the setback introduced with the difference in input and output length.

The model primarily consists of an encoder, a decoder, and an intermediate vector.

The encoder is a stack of various RNN units. The information drawn at each unit is forwarded to other units in the sequence. The hidden values $h_i$ can be depicted mathematically in Eq 7:

$$h_i = func\big(W^{hh}.h_{t-1} + W^{hx}.x_i\big) \tag{7}$$

Where 'W' is the appropriate weight applied.

The intermediate vector incorporates the information of all input elements. It is the final hidden state calculated by the encoder.

This is passed on to the Decoder which predicts the output for each step. It consists of various RNN units at each step that takes in the hidden state from the previous unit and generates an output along with its new hidden state that is further passed to the other units in the sequence. The output at each step is produced by the formula yi = softmax ($W^S$. ht) and the new hidden state is calculated using the given formula- hi = f($W^{hh}$. hi-1). The Encoder-Decoder architecture is shown in Fig 9.

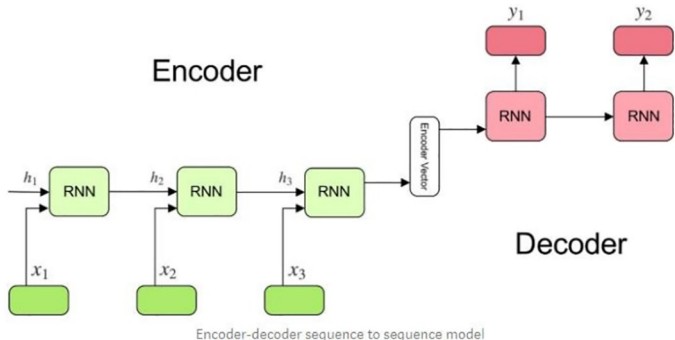

Encoder-decoder sequence to sequence model

**Fig 9. Encoder-decoder architecture.**

*3.2.1.1. Training data Preparation and Modeling.* In data preparation, each sentence is added with two tokens <sos> which is a start of sentence token, and <eos> for the end of a sentence. Each sentence is tokenized and padded with a maximum length. The sentences are passed to an embedding layer of 50 dimensions having pre-trained embedding weights. The output of the embedding layer is passed to encoder LSTM with a latent dimension of 200, the output of the encoder is passed to Decoder LSTM with same latent dimension. The output of decoder LSTM is further passed to Decoder embedding layer which gives an output of the sequence that is sampled using softmax sampling for final output. The dataset is divided into 80:20 ratios for training and validation.

**3.2.2. Modified attention mechanism in encoder decoder model with softmax overtime.** *3.2.2.1. Embedding layer.* Words in a document can be converted to their dense vector representation by utilizing word embedding. It is an efficient and better approach to the bag of words concept. Unlike the latter, the embedding layer works on the principle of creating a dense vector, and the position of the word is based on the words around it [23]. The embedding of a word is the position of that word in the vector space.

An embedding layer provided by Keras is used for bringing out the same information in text data for the neural network to process further. Global Vectors for Word Representation (GloVe), which is a pre-trained word embedding is used in this approach. The dataset on which the layer was trained consisted of almost 1billion words. It is available in different dimensions- 50, 100, 200, and 300. The Embedding layer from the pre-trained GloVe is used to seed for the words in this MELD dataset.

The layer of dimension 100 is used. This layer is applied after performing tokenization to convert text to a sequence. An example of the embedding in GloVe that contains a 100-dimension version in the ASCII format for the letter "the" has been shown below:

"the" - 0.038194–0.24487 0.72812–0.39961 0.083172 0.043953–0.39141 0.3344–0.57545 0.087459 0.28787–0.06731 0.30906–0.26384–0.13231–0.20757 0.33395–0.33848–0.31743–0.48336 0.1464–0.37304 0.34577 0.052041 0.44946–0.46971 0.02628–0.54155–0.15518–0.14107–0.039722 0.28277 0.14393 0.23464–0.31021 0.086173 0.20397 0.52624 0.17164–0.082378–0.71787–0.41531 0.20335–0.12763 0.41367 0.55187 0.57908–0.33477–0.36559–0.54857–0.062892 0.26584 0.30205 0.99775–0.80481–3.0243 0.01254–0.36942 2.2167 0.72201–0.24978 0.92136 0.034514 0.46745 1.1079–0.19358–0.074575 0.23353–0.052062–0.22044 0.057162–0.15806–0.30798–0.41625 0.37972 0.15006–0.53212–0.2055–1.2526 0.071624 0.70565 0.49744–0.42063 0.26148–1.538–0.30223–0.073438–0.28312 0.37104–0.25217 0.016215–0.017099–0.38984 0.87424–0.72569–0.51058–0.52028–0.1459 0.8278 0.27062.

Fig 10 shows the Encoder decoder attention Architecture. Attention mechanism further improves the efficiency and performance of the sequence to sequence model by overcoming the limitation of the encoder-decoder model to deal with long sequences. It behaves similar to the visual attention mechanism found in humans. The mechanism works by training the model to focus on specific elements. It pays attention to the important part of the instance. It does not generate a single intermediate vector as in the encoder- decoder model. This is done by giving weightage to time steps and including input from all units. This can better be understood by the statement that each output from a decoder is a weighted combination of all the input units. Also, it uses Gated Recurrent Unit (GRU) instead of LSTM and is implemented using bidirectional input.

In Eq 8, context vector which is the sum of the hidden states of the input sequence enables the decoder to focus on certain words that are responsible for predicting the output.

$$C_t = \sum_{i=1}^{n} \alpha_t, h_i \tag{8}$$

Next is to calculate the attention weights by using fully connected network and a Softmax function using Eq 9.

$$\alpha_{t,i} = align(y_t, x_i) \tag{9}$$

*3.2.2.2. Training Data Preparation and Modeling.* Data preparation and Encoder-Decoder architecture for attention mechanism model are the same as in Encoder- Decoder Model, where Encoder is now Bidirectional LSTM. An additional attention layer is added between encoder and decoder for calculating attention weights by using softmax over time sampling method. For preparing a custom attention layer following steps have been used.

1. Repeat vector layer which has a dimension of maximum length of the sentence.

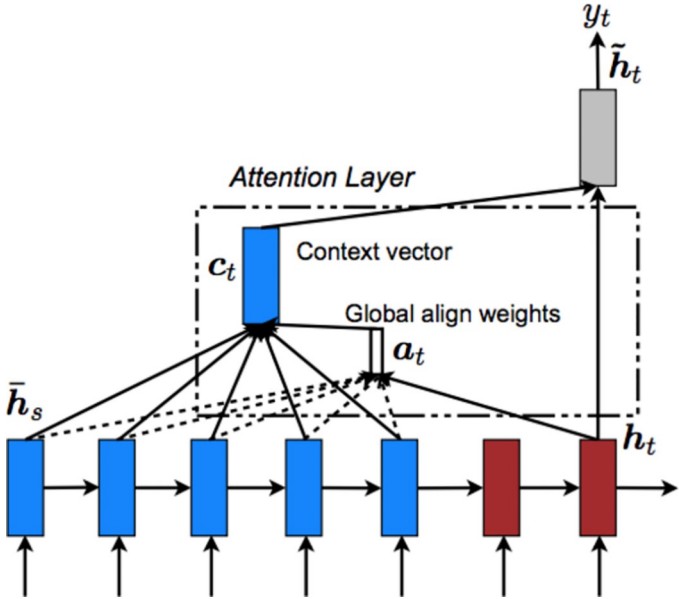

**Fig 10. Encoder decoder attention architecture [23].**

2. A concatenate layer concatenates the hidden state of the encoder that is h(t) and the previous timestamp input of the decoder that is s(t-1) where t is the time stamp or position of the token in a sequence.

3. Dense layer of 10 neurons with sigmoid activation.

4. The dense layer of 1 neuron with activation of custom softmax function which will give the probability of how much attention to pay to each word in an input sequence.

Above softmax will only calculate attention weights for a single hidden state but to make the softmax work properly, it needs to calculate the activation for all the hidden states in an encoder, such that the sum of all the attention weights is equal to one that gives a properly weighted average of hidden states. To accomplish this task, attention weights for all the hidden states are calculated at the same time, so that the final softmax has to be overall the sum of attention weights. All the hidden state values are passed to the same neural network to get all those outputs that can be used to calculate the softmax over time.

Eq 10 is used to calculate softmax overtime where out (t') is the output from the dense layer of 10 neurons.

$$\alpha(t') = \frac{expexp(out(t'))}{\sum_{t=1}^{Tx} exp(out(\tau))} \tag{10}$$

# 4. Results and discussion

## 4.1. Results

The customer satisfaction feedback survey is one of the crucial ones. In some situations, the voice based, or a facial expression-based survey feedback may lead to false opinion. So to overcome this weakness four various types of Deep language models such as: LSTM, BERT, Encoder Decoder and Attention Mechanism are implemented to classify the emotions of the customer feedback in text. In this section, the results of various models are given with performance comparison and the outcomes are discussed in detail. A total of four models are compared.

Fig 11 shows the LSTM based text classification. The graph depicts the relation between the epochs and the Acc/Loss. It clearly shows that, after training over 50 epochs', the accuracy was around 73% and loss was around 1.02. Since the accuracy was lesser, the predictions gave insignificant results.

Fig 12 shows the relation between Accuracy and the epochs. A pre trained BERT model is implemented. After fine-tuning the parameters such as setting Adam epsilon value to 1e-8. The figure shows the accuracy plot after training the BERT model over 50 epochs. The output predictions of the BERT model is shown in Fig 13, where if the input is "I just broke up with my boyfriend", the BERT model accurately predicted the emotion as "sad".

Fig 14 shows the model accuracy graph with respect to accuracy and epoch. For the encoder-decoder model, gives a better accuracy after training over 15 epochs. Fig 15 shows the prediction of the encoder-decoder model on smaller sentences, it can grasp words such as anger, love, hell, and then able to output the corresponding emotion value.

Fig 15 shows the prediction for the input statement is "I Hate You", the encoder decoder model accurately predicted the emotion as "anger".

Fig 16 shows the accuracy model graph for the attention mechanism model. It depicts that after training over 50 epochs the model gives a better accuracy. Some random sentences are taken from the validation set which the model has never seen before and tried the attention

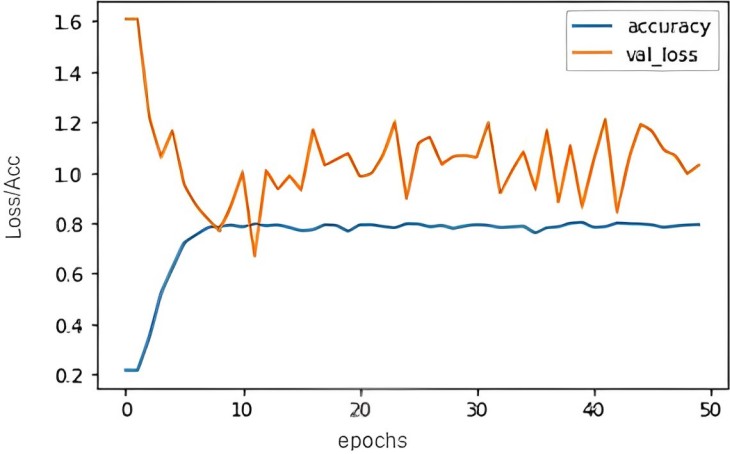

**Fig 11. LSTM Acc/Loss vs epoch plot.**

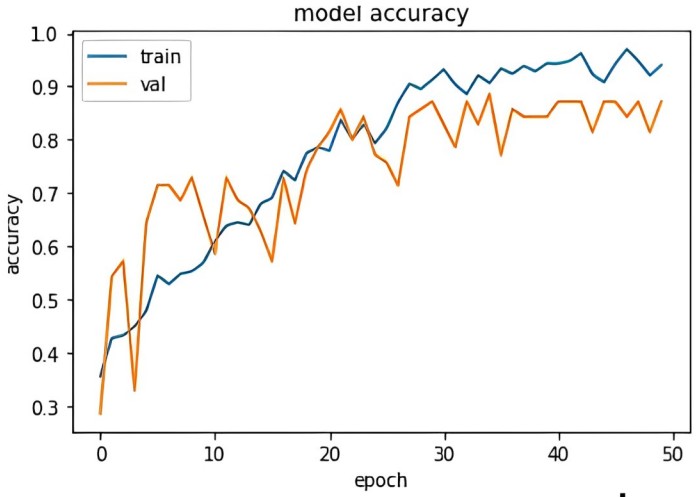

**Fig 12. BERT accuracy vs epoch plot.**

```
Sentence: I just broke up with my boyfriend
Predicton: sad
Sentence: Having passed the exam required to be an authorized Radio/TV dealer.
Predicton: joy
Sentence: Lucy sent him a resentful glance while adding an explanation for her presence
Predicton: anger
```

**Fig 13. BERT prediction.**

model for predictions. Fig 17 illustrates the prediction result. The model has identified the type of emotion on an unseen sentence.

## 4.2. Discussion

The core methodology of emotion classification algorithms in deep learning typically involves sequence classification, where each sequence is classified by the last layer of deep neural

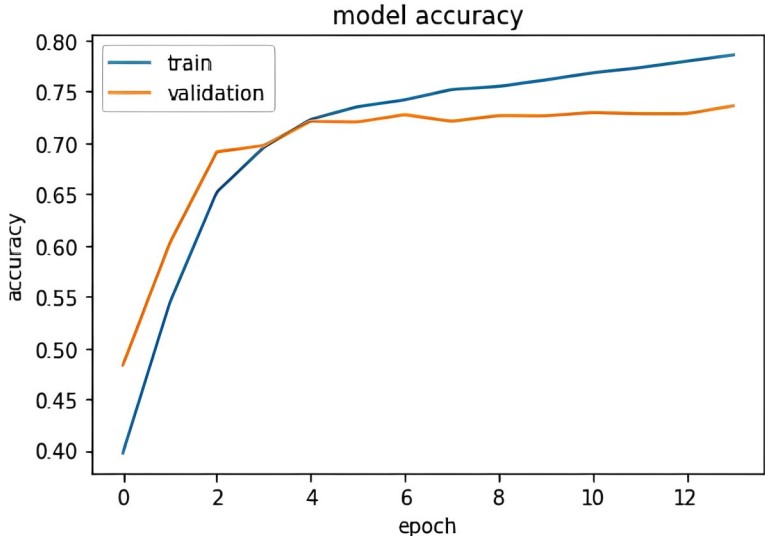

**Fig 14. Encoder decoder accuracy vs epoch plot.**

```
sentence: I Hate You
Prediction: anger
sentence: I love it!
Prediction: joy
sentence: What the hell is going on!
Prediction: anger
sentence: I am not happy
Prediction: neutral
```

**Fig 15. Encoder decoder prediction.**

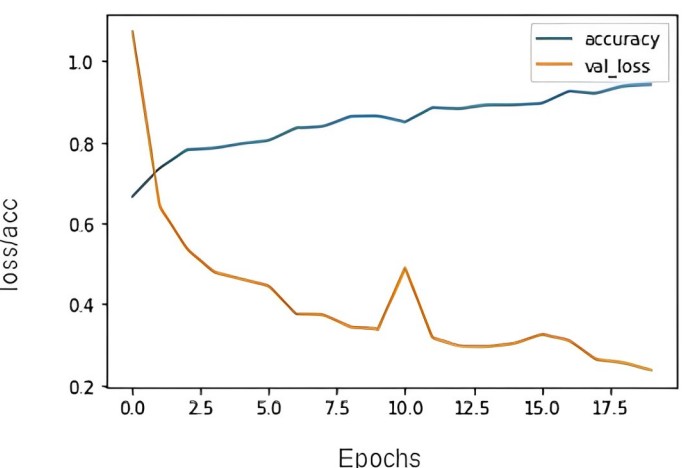

**Fig 16. Encoder decoder attention Acc/Loss vs epochs plot.**

```
Input sentence: gm friends its saturday happy fresh morning
Predicted translation: angry
Actual translation: <START> angry <END>
Continue? [Y/n]y

Input sentence: diego i know lo ame a jack
Predicted translation: neutral
Actual translation: <START> neutral <END>
Continue? [Y/n]y

Input sentence: if you re feeling the f omo will be hosting 2 more q as with the angry team o
Predicted translation: worry
Actual translation: <START> worry <END>
Continue? [Y/n]y

Input sentence: the dance video that haunts my dreams india is confused angry wtf more cheese than a bollywood d movie
Predicted translation: worry
Actual translation: <START> worry <END>
```

**Fig 17. Encoder decoder attention prediction.**

network into its type of the emotion. While this approach has shown promising results, it can encounter issues such as over fitting and inefficiency in processing large datasets, even after extensive training and parameter tuning. In contrast, our proposed model, which integrates a modified attention mechanism into an encoder-decoder framework, demonstrates significant improvements. Notably, it achieves state-of-the-art results with an impressive accuracy of 93.5% and an F1 score of 0.91 as shown in Table 3. This performance surpasses that of other methods like LSTM (73.22% accuracy), BERT (84.1% accuracy), and traditional Encoder-Decoder models (73.18% accuracy).

There is no research work to compare the exact results of the proposed model, since we have proposed a **modified** encoder decoder model with attention mechanism, still a comparison is presented in Table 4 with the closely related works on text based emotion recognition.

**The major strengths of the proposed method are**;

- Enhanced Accuracy and Precision: By incorporating softmax over time to calculate attention weights for each word, our model achieves higher accuracy and F1 scores.

- Improved Handling of Time Delays and Gap length: The model effectively decodes sequences with variable time delays, an area where LSTM also performs well but with comparatively less accuracy. LSTMs are also relatively insensitive to gap lengths in sequences.

**The major challenges of the proposed method are**;

- Complexity and Computational Demand: The modified attention mechanism, while effective, adds complexity to the model, potentially increasing the computational resources required.

- Limited Generalizability: The current implementation may be optimized for specific types of emotional sequence data, raising questions about its generalizability across diverse dataset.

In conclusion, while our model sets a new benchmark in emotion classification with its advanced attention mechanism, considerations around its complexity, risk of overfitting, and generalizability must be addressed in future research.

**Table 3. Performance comparison of various models.**

| S.No | Model | Validation Accuracy (%) | F1 Score |
|---|---|---|---|
| 1 | LSTM | 73.22 | 0.71 |
| 2 | BERT | 84.1 | 0.85 |
| 3 | Encoder decoder | 73.18 | 0.75 |
| 4 | Attention mechanism | **93.5** | **0.91** |

**Table 4. Comparison of various algorithms on various dataset.**

| Paper | Model used | Dataset | Accuracy | F1score |
|---|---|---|---|---|
| Sattar B. Sadkhan et al [10] | Fuzzy Logic | | >85% | |
| Meena G et al [12] | CNN-based inception-3 | JAFFE | 86 | 44 |
| | | FER2013 | 73.09 | 38 |
| Meena G et al [14] | XceptionNet | FER2013 | 77.92 | 57 |
| | InspectionV3 | | 73.09 | 38 |
| | VGG-19 | | 65.41 | 62 |
| | XceptionNet | JAFFE | 92.50 | 6 |
| | InspectionV3 | | 86 | 44 |
| | VGG-19 | | 94.00 | 95 |
| Proposed Model | LSTM | MELD, ISEAR and We Feel Fine | 73.22 | 0.71 |
| | BERT | | 84.1 | 0.85 |
| | Encoder decoder | | 73.18 | 0.75 |
| | Attention mechanism | | 93.5 | 0.91 |

## 4.3. Future scope

Emotion classification from speech signals is growing in multi directions each day. It has wide applications in customer satisfaction surveys, Audience feedback system, tele-consultation system, tele counselling systems. It has even a scope of saving lives by capturing the emotions of the person whose speech signal is under study. By attention mechanism with the text weighing process, not only the speech features but also the content of the speech are used to understand the emotion. The main objective of this work is to find the "True" emotions from the speech signal.

Building on this foundation, future research should explore the application of this technology across diverse settings, further enhancing its adaptability and usefulness. Key areas for development include refining the model to capture a broader spectrum of emotional expressions, which may vary greatly across different individuals and cultures. Additionally, extending the model to recognize and interpret a wider range of languages will significantly broaden its applicability and impact. This inclusive approach is particularly important in understanding and helping individuals with speech disabilities, also ensuring that the system is not just foolproof, but also universally accessible and effective in detecting true emotions in speech and content data across various languages and cultural contexts.

Our approach can be extended and applied across different sectors, impacting various aspects of daily life and professional environments. In telemedicine and psychiatric care, understanding emotional cues can help in the diagnosis and treatment of mental health disorders. Speech emotion recognition can also be used in monitoring the emotional well-being of patients with chronic illnesses. Another such application or extension of this research can be in the Education industry where this technology can help teachers understand students' engagement and emotional responses to learning materials which can help in enabling personalized education experiences. This research also has applicability in the automotive industry wherein smart vehicles can detect drivers' state of mind to ensure safety. For instance, detecting stress or anger could trigger calming music or a suggestion to take a break or enable a safe driving mode with controlled speed. The AI industry can also benefit from this research by enhancing the virtual assistants with emotion recognition capabilities to make interactions more human-like and responsive to the user's emotional state.

## 5. Conclusion

Recognizing true emotions from speech has wide applications and is a significant challenge, especially since feature-based emotion classifiers can sometimes be misled by false tones, leading to incorrect spectral characteristics. This research has advanced our ability to identify the true type of emotion in text, taking us closer to more accurate emotion recognition. The main objective of this study was to develop a deep model capable of capturing voice signals, processing them, and then dividing the task into two parallel models. The first model detects the type of emotion in speech [24], while the second converts the voice into text, preprocesses the text, and converts it into a vector for a modified attention mechanism with a softmax over time model to classify the emotion.

The implemented models, trained over multiple epochs with different approaches like LSTM, BERT, encoder-decoder models, and attention mechanisms, have shown promising results. The LSTM model achieved an accuracy of around 73% after 50 epochs, while the BERT model, fine-tuned with specific parameters, also demonstrated significant accuracy. The encoder-decoder model, trained over 15 epochs, achieved similar accuracy but with a reduced loss value. The attention mechanism, after 50 epochs of training, reached an impressive accuracy of 93.55% with a low loss of 0.2.

The findings of this research, particularly the success of the attention-based algorithm, indicate strong potential in the field of emotion recognition from voice as well as text forms. This model contributes directly to customer feedback mechanisms and similar applications to recognize true feedback from spoken or written text. The results clearly demonstrate that the implemented attention-based model holds significant promise for future research in this area.

## Acknowledgments

The authors thank Vellore Institute of Technology, Vellore for providing technical support and the resource support to carry out this research work.

## Author Contributions

**Conceptualization:** Elena Lyakso, Nersisson Ruban.

**Data curation:** Megha Roshan.

**Formal analysis:** Elena Lyakso.

**Investigation:** Mukul Rawat, Nersisson Ruban.

**Methodology:** Megha Roshan, Nersisson Ruban.

**Project administration:** A. Mary Mekala.

**Software:** Megha Roshan, Karan Aryan.

**Supervision:** Elena Lyakso.

**Validation:** Mukul Rawat, Karan Aryan, A. Mary Mekala.

**Writing – original draft:** Megha Roshan, Mukul Rawat, Karan Aryan.

**Writing – review & editing:** A. Mary Mekala, Nersisson Ruban.

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
