## [Decision Letter · Decision Letter 0]

10 Dec 2023

PONE-D-23-22091Linguistic based Emotions Analysis Using Softmax over Time Attention MechanismPLOS ONE

Dear Dr. Ruban,

Thank you for submitting your manuscript to PLOS ONE. After careful consideration, we feel that it has merit but does not fully meet PLOS ONE’s publication criteria as it currently stands. Therefore, we invite you to submit a revised version of the manuscript that addresses the points raised during the review process.

We look forward to receiving your revised manuscript.

Kind regards,

Michal Ptaszynski, PhD

Academic Editor

PLOS ONE

Journal Requirements:

"The authors thank Vellore Institute of Technology, Vellore for providing ‘VIT SEED GRANT’ for carrying out the initial study related to this research work."

"The authors received no specific funding for this work"

Reviewers' comments:

Reviewer's Responses to Questions

**Comments to the Author**

1. Is the manuscript technically sound, and do the data support the conclusions?

Reviewer #1: Partly

Reviewer #2: Yes

Reviewer #3: Yes

2. Has the statistical analysis been performed appropriately and rigorously? 

Reviewer #1: Yes

Reviewer #2: N/A

Reviewer #3: Yes

3. Have the authors made all data underlying the findings in their manuscript fully available?

Reviewer #1: No

Reviewer #2: Yes

Reviewer #3: Yes

4. Is the manuscript presented in an intelligible fashion and written in standard English?

Reviewer #1: Yes

Reviewer #2: No

Reviewer #3: Yes

5. Review Comments to the Author

Reviewer #1: I read this manuscript over and over again but I failed to find what was the state-of-the-art of this research. The title says attention mechanism, but in the background it doesn't discuss why this method is proposed

Multimodal classification, namely voice and textual input, also does not explain the results. Fig. 1 breaks down the two classifications, then how to vote on the classification results.

I think this manuscript fails to convey the urgency of the proposed research. This can be seen because there is no emphasis on what the author did and what his contribution was in the introduction

I don't see the benefit of giving colors to Figures, for example Fig. 2 to Fig. 5. To describe the distribution of a dataset, just use a regular picture or a table of numbers

Reviewer #2: - The introduction should clarify: What are major research objectives? Are there any existing solutions? Which is the best? What is its main limitation? What does the author hope to achieve?

- Also include the problem identification and research gaps.

- The authors should highlight the manuscript's innovations and contributions.

- Please make the Introduction and related work sections more productive using the following articles. Reading and using these articles and also cited in this article: Identifying emotions from facial expressions using a deep convolutional neural network-based approach. Multimedia Tools and Applications (2023): 1-22.; Facial emotion recognition and music recommendation system using CNN-based deep learning techniques. Evolving Systems (2023): 1-18.; Sentiment analysis on images using convolutional neural networks-based Inception-V3 transfer learning approach. International Journal of Information Management Data Insights 3.1 (2023): 100174.; Sentiment analysis on images using different transfer learning models. Procedia Computer Science, 218, 1640-1649. ; Image-based sentiment analysis using InceptionV3 transfer learning approach." SN Computer Science 4.3 (2023): 242.

- Results may be describe in proper way.

- Experimental details, simulation environment, measuring parameters etc. need to be added.

- Author may compare their work with existing literature.

- All figures have low quality, and please improve all of them.

- Improve discussion section. It must include the strengths and weaknesses of the proposed approach.

- Expand the critical results in the conclusion. Focus on the main developments in the finale. Also, write the main contributions in the conclusion.

- The grammatical error should be corrected.

Reviewer #3: This paper presents an innovative approach to emotion recognition, focusing on textual data analysis to complement the traditional methods that rely on speech signal features. The authors rightly identify a significant gap in current emotion detection systems, which often misinterpret emotions due to limitations in analyzing speech frequency and pitch, or can be misled by false facial expressions in video-based systems.

The proposed solution, leveraging a modified Encoder-Decoder model with an attention mechanism, is a notable advancement. The authors' approach to integrate text-based emotion recognition into the feedback management process is both novel and practical. This integration could indeed enhance the accuracy of emotion detection in various applications, particularly in customer feedback and medical scenarios.

The achievement of a 93.5% accuracy rate with the modified text-based classification model is impressive. This statistic suggests a significant improvement over existing models and indicates the potential of this method in real-world applications. However, the paper would benefit from a more detailed discussion of the datasets used for training and validation, as well as the specific types of emotions that were recognized and classified.

Additionally, the paper might be strengthened by addressing potential limitations or biases in text-based emotion recognition, such as cultural or linguistic variations that could impact the model's effectiveness. Exploring the scalability of the model and its applicability across different domains would also be valuable.

In conclusion, the research presents a promising direction in emotion recognition technology. The integration of text-based analysis using a sophisticated attention mechanism offers a more nuanced and potentially more accurate method for emotion detection. Future research could build on this work by exploring its application in diverse settings and further refining the model to account for a wider range of emotional expressions and languages.

6. PLOS authors have the option to publish the peer review history of their article (what does this mean?). If published, this will include your full peer review and any attached files.

Reviewer #1: **Yes: **Dewi Yanti Liliana

Reviewer #2: No

Reviewer #3: No

---

## [Author Response · Author response to Decision Letter 0]

30 Jan 2024

Linguistic based Emotions Analysis Using Softmax over Time Attention Mechanism

Responses to the Reviewers' comments:

Authors would like to sincerely appreciate the reviewer’s valuable comments for the improvement of our manuscript. We have revised the manuscript based on the comments received from the reviewers, and modified the manuscript to best suit the standards of the journal. We are ready for any more revisions also for the betterment of the paper.

 The authors state that there is no conflict of interest with the suggested reviewers. More over the authors wants to express their heart felt greetings to the reviewers for their valuable suggestion for the improvement of this work. 

 We have incorporated all the suggestions given by the reviewers and the corrections made in the manuscript are highlighted. We have also given detailed response for the comments given by the reviewers below.

Reviewer #1: 

1. I read this manuscript over and over again but I failed to find what was the state-of-the-art of this research. The title says attention mechanism, but in the background it doesn't discuss why this method is proposed

Response: We thank the reviewer for this observation, the state of the art of this research is that, the softmax function independently applies to each set of logits (Logits are the outputs of a neural network before the activation function is applied). This converts the logits into a probability distribution for each emotion category at each time step. This means that for every word or token in the text, you have a probability distribution over emotions. This explanation is included in the manuscript as suggested by the reviewer in Page 9 under 2.3 algorithms.

More over we applied the attention mechanism to the encoder decoder model which was never tried before for text based emotion recognition system. Converting the words in the statement into their dense vector layer is the basis for the attention mechanism and incorporating it into the encoded decoded concept to get a better performance is the major contribution of this research. This is discussed in the section 3.2.2. 

2. Multimodal classification, namely voice and textual input, also does not explain the results. Fig. 1 breaks down the two classifications, then how to vote on the classification results.

Response: We thank the reviewer for this observation, to obtain a final emotion label, we combines the outcomes from both modalities, taking into account their respective confidence scores and output the resulted emotion which has highest confidence score. The mechanism of choosing the highest confidence level is used as the voting scheme in the proposed approach. This multimodal approach enhances the overall accuracy and reliability of our emotion classification system by harnessing complementary information from both voice and text sources.

3. I think this manuscript fails to convey the urgency of the proposed research. This can be seen because there is no emphasis on what the author did and what his contribution was in the introduction.

Response: We thank the reviewer for this observation; this research highlights the pressing need for more robust emotion recognition systems and underscores the potential of transfer models with attention mechanisms to significantly improve feedback management processes and the medical applications.

The major technical contribution of the authors are briefly included in the introduction section as suggested by the reviewer and it is elaborated in other sections of the manuscript.

4. I don't see the benefit of giving colors to Figures, for example Fig. 2 to Fig. 5. To describe the distribution of a dataset, just use a regular picture or a table of numbers

Response: We thank the reviewer for this observation; there is no technical reason for the color images, it was given to make a good look of the article, since it was suggested by the reviewer, we have modified the images into grey scale images.

Reviewer #2: -

1. The introduction should clarify: What are major research objectives? Are there any existing solutions? Which is the best? What is its main limitation? What does the author hope to achieve?

- Also include the problem identification and research gaps.

- The authors should highlight the manuscript's innovations and contributions.

- Please make the Introduction and related work sections more productive using the following articles. Reading and using these articles and also cited in this article: Identifying emotions from facial expressions using a deep convolutional neural network-based approach. Multimedia Tools and Applications (2023): 1-22.; Facial emotion recognition and music recommendation system using CNN-based deep learning techniques. Evolving Systems (2023): 1-18.; Sentiment analysis on images using convolutional neural networks-based Inception-V3 transfer learning approach. International Journal of Information Management Data Insights 3.1 (2023): 100174.; Sentiment analysis on images using different transfer learning models. Procedia Computer Science, 218, 1640-1649. ; Image-based sentiment analysis using InceptionV3 transfer learning approach." SN Computer Science 4.3 (2023): 242.

Response: We thank the reviewer for the valuable comment, as per the suggestion we have included the contribution, innovation in the article (Introduction section, section 2.3). We have highlighted the objective and included the advantage and limitations in the conclusion section of the manuscript. 

Thank you for suggesting related articles, we find it is very much related to the proposed research in the application context, so we have referred those articles and cited in the appropriate places in the article.

2. Results may be describe in proper way.

Response: As per the suggestion, we have completely rewritten the result section and included more discussion on various plots and prediction given in the result section.

3. Experimental details, simulation environment, measuring parameters etc. need to be added.

Response: We designed a versatile simulation environment having Dataset from different diversity, culture, age groups as well as gender. This simulation framework allowed us to replicate real-world scenarios, introducing controlled variations in cultural context, linguistic diversity, and emotional expressions. By simulating diverse conversational dynamics and cross-cultural variations, we assessed the adaptability of our models to different contexts. This simulation approach played a pivotal role in uncovering potential limitations, biases, and challenges that may arise due to cultural, linguistic, and contextual differences, ensuring the robustness and applicability of our models across various real-world scenarios. Biasness can be improved by fine tuning the same model with a new dataset in different simulation environment.

4. Author may compare their work with existing literature.

Response: We have introduced three dataset for textual emotion classification, which is very rarely used by any other research group and moreover we have proposed an algorithm which uses dense vector of the words in the sentences to predict the emotions, which is completely a new concept and the state-of-the art approach, we could not find close researches, still we have compared the existing research work which are more close to the proposed research work. The performance comparison is included as Table-4

5. All figures have low quality, and please improve all of them.

Response: As per the suggestion, all the plots which have poor resolution was improved and modified into a good resolution images.

6. Improve discussion section. It must include the strengths and weaknesses of the proposed approach.

Response: As per the suggestion, the discussion section is completely rewritten and the strengths of the proposed methodology and the challenges are discussed in the revised discussion section.

7. Expand the critical results in the conclusion. Focus on the main developments in the finale. Also, write the main contributions in the conclusion.

Response: As per the suggestion, the conclusion section is rewritten, the revised version includes, future directions, major focus on the application and the technical contributions of the research.

8. The grammatical error should be corrected.

Response: The whole article is checked for the grammatical error and the language check is performed, and we feel that the article is much improved in the language and grammar context.

Reviewer #3: 

This paper presents an innovative approach to emotion recognition, focusing on textual data analysis to complement the traditional methods that rely on speech signal features. The authors rightly identify a significant gap in current emotion detection systems, which often misinterpret emotions due to limitations in analyzing speech frequency and pitch, or can be misled by false facial expressions in video-based systems.

The proposed solution, leveraging a modified Encoder-Decoder model with an attention mechanism, is a notable advancement. The authors' approach to integrate text-based emotion recognition into the feedback management process is both novel and practical. This integration could indeed enhance the accuracy of emotion detection in various applications, particularly in customer feedback and medical scenarios.

The achievement of a 93.5% accuracy rate with the modified text-based classification model is impressive. This statistic suggests a significant improvement over existing models and indicates the potential of this method in real-world applications. However, the paper would benefit from a more detailed discussion of the datasets used for training and validation, as well as the specific types of emotions that were recognized and classified.

1. Additionally, the paper might be strengthened by addressing potential limitations or biases in text-based emotion recognition, such as cultural or linguistic variations that could impact the model's effectiveness. Exploring the scalability of the model and its applicability across different domains would also be valuable.

Response: Dear Reviewer, we sincerely thank you for your appreciation for our research work, it gives us great motivation to do more research in the domain, as per your suggestion we have included about the cultural diversity and various biases in our manuscript. This inclusion is made in section 2: materials and methods. We are giving the content below for your quick reference:

‘We designed a versatile simulation environment having Dataset from different diversity, culture, age groups as well as gender. This simulation framework allowed us to replicate real-world scenarios, introducing controlled variations in cultural context, linguistic diversity, and emotional expressions. By simulating diverse conversational dynamics and cross-cultural variations, we assessed the adaptability of our models to different contexts. This simulation approach played a pivotal role in uncovering potential limitations, biases, and challenges that may arise due to cultural, linguistic, and contextual differences, ensuring the robustness and applicability of our models across various real-world scenarios. Biasness can be improved by fine tuning the same model with a new dataset in different simulation environment.’

In conclusion, the research presents a promising direction in emotion recognition technology. The integration of text-based analysis using a sophisticated attention mechanism offers a more nuanced and potentially more accurate method for emotion detection. 

2. Future research could build on this work by exploring its application in diverse settings and further refining the model to account for a wider range of emotional expressions and languages.

Response: Yes, there is an enormous scope for this kind of textual emotion recognition system with the fake emotion proof, as per the suggestion we have included a separate future scope section in the conclusion which gives an overview of all the possible future directions of this research work.

---

## [Decision Letter · Decision Letter 1]

14 Mar 2024

Linguistic based Emotions Analysis Using Softmax over Time Attention Mechanism

PONE-D-23-22091R1

Dear Dr. Ruban,

We’re pleased to inform you that your manuscript has been judged scientifically suitable for publication and will be formally accepted for publication once it meets all outstanding technical requirements.

Kind regards,

Michal Ptaszynski, PhD

Academic Editor

PLOS ONE

Additional Editor Comments (optional):

Reviewers' comments:

Reviewer's Responses to Questions

**Comments to the Author**

1. If the authors have adequately addressed your comments raised in a previous round of review and you feel that this manuscript is now acceptable for publication, you may indicate that here to bypass the “Comments to the Author” section, enter your conflict of interest statement in the “Confidential to Editor” section, and submit your "Accept" recommendation.

Reviewer #1: All comments have been addressed

Reviewer #2: All comments have been addressed

2. Is the manuscript technically sound, and do the data support the conclusions?

Reviewer #1: Yes

Reviewer #2: Yes

3. Has the statistical analysis been performed appropriately and rigorously? 

Reviewer #1: N/A

Reviewer #2: N/A

4. Have the authors made all data underlying the findings in their manuscript fully available?

Reviewer #1: Yes

Reviewer #2: Yes

5. Is the manuscript presented in an intelligible fashion and written in standard English?

Reviewer #1: Yes

Reviewer #2: Yes

6. Review Comments to the Author

Reviewer #1: The author has added state-of-the-art contributions to the abstract. The author has added a sentence on how to vote on emotion classification results in a multimodal system. The author has added how to calculate the softmax function for each emotion class using a probability distribution.

Reviewer #2: (No Response)

7. PLOS authors have the option to publish the peer review history of their article (what does this mean?). If published, this will include your full peer review and any attached files.

Reviewer #1: **Yes: **Dewi Yanti Liliana

Reviewer #2: No

---

## [Editor Report · Acceptance letter]

3 Apr 2024

PONE-D-23-22091R1 

PLOS ONE

Dear Dr. Ruban, 

I'm pleased to inform you that your manuscript has been deemed suitable for publication in PLOS ONE. Congratulations! Your manuscript is now being handed over to our production team.

Kind regards, 

on behalf of

Dr. Michal Ptaszynski 

Academic Editor

PLOS ONE